# Gradient-based Training of Slow Feature Analysis by Differentiable Approximate Whitening

## Abstract

We propose Power Slow Feature Analysis, a gradient-based method to extract temporally slow features from a high-dimensional input stream that varies on a faster time-scale, as a variant of Slow Feature Analysis (SFA). While displaying performance comparable to hierarchical extensions to the SFA algorithm, such as Hierarchical Slow Feature Analysis, for a small number of output-features, our algorithm allows fully differentiable end-to-end training of arbitrary differentiable approximators (e.g., deep neural networks). We provide experimental evidence that PowerSFA is able to extract meaningful and informative low-dimensional features in the case of (a) synthetic low-dimensional data, (b) visual data, and also for (c) a general dataset for which symmetric non-temporal relations between points can be defined.

## 1 Introduction

Finding meaningful representations in data is a core challenge in modern machine learning as the performance in many goal-directed frameworks, such as reinforcement learning or supervised learning, is strongly influenced by the quality of the former. Usually, features are either domain specific or acquired through learning. Most currently successful approaches for either deep supervised learning (Goodfellow et al. (2016)) or reinforcement learning (Sutton & Barto (1998)) rely on a training signal, i.e., a classification label or reward, to provide sufficient indication which features of the input data should be extracted to increase performance. However, in most real-world scenarios labels have to be provided by expert knowledge and reward signals are sparse. In unsupervised representation learning, one tries to find and apply a principle by which to extract meaning from data without assuming the availability of any goal-driven metrics.

Examples for such principles are based on reconstruction error in principal component analysis or autoencoder networks (Bourlard & Kamp (1988)), statistical dependence of extracted features in independent component analysis (Comon (1994)), indistinguishability of synthetically generated data from samples of the input distribution in generative adversarial nets (Goodfellow et al. (2014)). Further examples include: fitting the probability distribution of input data with variational autoencoders (Kingma & Welling (2013)), neighborhood preservation used in locally linear embedding (LLE, Roweis & Saul (2000)), Laplacian eigenmaps (LEM, Belkin & Niyogi (2003)), as well as temporal coherence as used in slow feature analysis, (SFA, Wiskott & Sejnowski (2002)) or regularized slowness optimization (Bengio & Bergstra (2009)). Temporal coherence is the focus of this work and has been shown to provide a useful proxy for extracting underlying causes from time-series data such as position, head direction, identity of spatial view similar to those observed in rodent brains (Franzius et al. (2007)) from ego-visual data. The principle has also been successfully applied to determining object configuration (identity, position, angle) from a tabletop view of moving objects (Franzius et al. (2011)) in the form of SFA. A graph-based generalization of SFA has been used to achieve (at that time) state-of-the-art age estimation results on the MORPH dataset in Escalante-B. (2017).

We propose a variant of SFA that approximately enforces the same constraints while being differentiable and thus allows training by gradient-descent-derived methods. This makes it possible to leverage the representational power of complex models, such as deep neural networks, and useful

ideas from that domain (dropout, batch normalization, activation regularization) to extract slow and informative features from data. To demonstrate the applicability of this approach, we provide three distinct experimental evaluations.

## 2 RELATED WORK

### 2.1 SLOWNESS-BASED METHODS

While the original proposal of SFA by Wiskott & Sejnowski (2002) uses non-linear basis functions as a method to introduce non-linearity to the otherwise linear model, this classical approach has two limitations: **(a)** The basis functions are fixed and thus have to be chosen beforehand by expert knowledge or trial-and-error, and **(b)** as the resulting model is shallow, its expressivity tends to scale unfavorably in the dimension of the expansion (Raghu et al. (2016)) compared to hierarchical extensions discussed later. In the case of polynomial expansion, expanding to degree $d$ on $e$-dimensional input data results in $\binom{d+e}{d}$-dimensional expanded data, e.g., quadratically expanding grayscale images of $180 \times 90$ pixels results in an output dimension $> 131 \cdot 10^6$. Extracting a single feature with a linear model would thus require 5.7-times more parameters than the modern and powerful *Xception* network (Chollet (2016)).

An alternative approach to increase expressivity using expansions is to apply low-degree non-linearities repeatedly in a receptive-field fashion, interlacing them with projection steps. This has been done in Hierarchical SFA (HSFA, Franzius et al. (2007) and Escalante-B. & Wiskott (2016)) and deep architectures in general. But while the latter are typically trained in an end-to-end fashion by variants of stochastic gradient-descent, HSFA is trained in a layer-wise procedure, solving the linear SFA problem consecutively for each layer as closed-form solution and thereby assuming that globally slow features can be composed of decreasingly local slow features. Escalante-B. & Wiskott (2016) shows that this assumption can be partially relaxed by adding information by-passes to the model. Compared to HSFA, our method allows directly modifying parameters in different layers to optimize a global slowness objective.

Our algorithm is in line with recent work harnessing the temporal coherence prior by Bengio et al. (2013) in deep, self-supervised feature learning. This can be done by having a slowness term in the loss function. To avoid the trivial, constant solution, another term is usually added. For example, a reconstruction loss in an auto-encoder (Goroshin et al. (2015a)) or one-step latent code prediction (Goroshin et al. (2015b)). Deep temporal coherence has also been considered via the lens of similarity metric learning, for example by optimizing a contrastive loss (Jayaraman & Grauman (2016), Mobahi et al. (2009)) or a triplet loss (Jansen et al. (2017), Wang & Gupta (2015)). These metric learning approaches manage to avoid degenerate solutions by pushing points away from each other (in feature space) that are not temporal neighbors. Our work differs from these approaches as we seek to directly approximate the optimization problem as originally stated by Wiskott & Sejnowski (2002) in a deep learning setting, automatically handling the constant solution and ensuring that different features code for different information.

### 2.2 GRAPH-BASED METHODS

SFA has been generalized to *graph-based SFA* (GSFA, Escalante-B. & Wiskott (2013)) and *generalized SFA* (Sprekeler (2011)). The former adapts only the objective function, while the latter additionally generalizes the constraints to **D**-orthogonality (**D** being the degree matrix of an underlying graph). Following generalized SFA, standard SFA can be shown to be a special case of Laplacian Eigenmaps.

Spectral Inference Networks (SpIN, Pfau et al. (2018)) utilize this connection to successfully derive a gradient-based SFA training as a special case. They are based on correcting a biased gradient when directly optimizing the Rayleigh-Quotient with respect to the model's parameters. The constraints are implicitly enforced through the loss function as opposed to directly whitening the output. Similar to our work, SpINs allow for employing any architecture to find these embeddings. However, the whitening proposed in this paper is applicable to any loss function as it is part of the model architecture and not inherently a part of the optimized objective.

The generalization of PowerSFA proposed in section 6.3 is close to GSFA, assuming regularity (or rather: ignoring non-regularity) in the graph for the orthogonality constraint.

SpectralNets (SN, Shaham et al. (2018)) are another closely related approach in which a differentiable approximator is trained to learn spectral embeddings used in subsequent $k$-means clustering. Opposed to PowerSFA and SpINs, they split a single optimization step into two parts: an ortho-normalization step based on explicitly calculating the Cholesky decomposition of the batch covariance matrix to set and freeze the weights of a linear output layer, followed by a stochastic optimization step. While this can be considered end-to-end depending on the paradigm, Shaham et al. (2018) do not indicate if and how this can be implemented as fully differentiable architecture.

## 2.3 OTHER RELATED APPROACHES

In section 6.3 we consider a generalization of PowerSFA's loss similar to GSFA and apply it to the NORB dataset (LeCun et al. (2004)). We thereby loosely follow the experimental procedure in Hadsell et al. (2006). The authors use a siamese neural network architecture (Bromley et al. (1993)) for optimizing pair-wise distances of embedded points to reflect similarity and dissimilarity structure of the data. In particular, they do not enforce orthogonality of the embeddings but rely on the optimization procedure to maximize informativeness.

## 3 SLOW FEATURE ANALYSIS

Slow Feature Analysis (SFA) is based on the hypothesis that interesting high-dimensional streams of data that vary quickly in time, are typically caused by a low number of underlying factors that vary comparably slow. Therefore, slowness can be used as a proxy criterion by which to extract meaningful representations of these low-dimensional underlying causes, even in the absence of labels.

There is strong evidence in favor of this hypothesis, as it has been shown that features extracted by SFA tend to encode highly relevant information about the data-generating environments, e.g., slow features encode and disentangle object identity, rotation, and position in visual tasks (Franzius et al. (2011)) as well as agent position and orientation from visual first-person recordings of random movement similar to place cells in rodents, or head-direction cells in primates (Franzius et al. (2007)).

The notion of extracting slow features from a time-series dataset has been formalized as a sequential optimization problem. Given a time-series $\{x_t\}_{t=0...\tau}$ with $x_t \in \mathbb{R}^d$, sequentially find continuous functions $g_i : \mathbb{R}^d \to \mathbb{R}$ with:

$$\min_{g_i} \quad \left\langle (g_i(x_{t+1}) - g_i(x_t))^2 \right\rangle_t \tag{1a}$$

$$\text{s.t.} \quad \left\langle g_i(x_t) \right\rangle_t = 0, \tag{1b}$$

$$\left\langle g_i(x_t)^2 \right\rangle_t = 1, \tag{1c}$$

$$\left\langle g_i(x_t) g_j(x_t) \right\rangle_t = 0, \quad \forall j < i \tag{1d}$$

where $\left\langle \cdot \right\rangle_t$ is the time-average. The constraints ensure that each of the extracted features is informative (decorrelated to all others, equation (1d)) and non-trivial (unit variance, equation (1c)). Originally, solutions to SFA directly were only proposed for the space of affine functions $g_i \in \mathcal{G}$, for which a closed-form solution exists, and in a kernelized version that requires strong regularization.

## 4 (APPROXIMATE) WHITENING

The standard implementation of linear SFA (Zito et al. (2008)) is based on computing the closed-form solution to a generalized eigenvalue problem (Berkes & Wiskott (2005)). Another approach is to first whiten the time-series data, followed by a projection onto the minor components of the difference time-series $\{\dot{x}_t = x_{t+1} - x_t\}_{t=0...\tau-1}$.

Whitened data has three important properties: **(a)** it is mean-free (constraint (1b)), **(b)** has unit variance if projected onto an arbitrary unit vector (constraint (1c)), and **(c)** projections onto orthonormal vectors are decorrelated (constraint (1d)).

For a dataset $\tilde{\mathbf{X}} \in \mathbb{R}^{d \times N}$ with $N$ and $d$ being size and dimension of the dataset, respectively, the corresponding whitened dataset is defined as

$$\mathbf{X} = \mathbf{W}\tilde{\mathbf{X}}$$

with $\mathbf{W} = \mathbf{D}^{-\frac{1}{2}}\mathbf{U}^T$ and $\mathbf{C} = \mathbf{U}\mathbf{D}\mathbf{U}^T \in \mathbb{R}^{d \times d}$ being the whitening matrix and the canonical eigendecomposition of the covariance matrix, respectively. $\mathbf{U}$'s columns contain the eigenvectors of $\mathbf{C}$ and $\mathbf{D}$ contains the corresponding eigenvalues on its diagonal. As $\mathbf{C}$ is typically assumed to be positive definite, $\mathbf{D}^{-\frac{1}{2}}$ is well-defined.

One widely used method to extract eigenvector/-value pairs is **power iteration**. Starting from a random vector $u_0 \in_R \mathbb{R}^d$, repeatedly applying:

$$u_{i+1} = \frac{\mathbf{C}u_i}{\|\mathbf{C}u_i\|}$$

converges to the eigenvector $u$ corresponding to the largest (absolute) eigenvalue $\lambda$. The eigenvalue can then be extracted as

$$\lambda = \|\mathbf{C}u\|$$

and the spectral component corresponding to this eigenvector can be removed as

$$\mathbf{C} \leftarrow \mathbf{C} - \lambda \mathbf{u}\mathbf{u}^T.$$

Repeating the procedure converges to the eigenvector corresponding to the next largest eigenvalue and so on. We use a previously fixed number of iterations for this method as experiments have shown that approximate whitening is enough to take meaningful optimization steps. In practice, a relatively small number of iterations results in acceptable whitening for most non-degenerate cases.

---

**Data:** covariance matrix $\mathbf{C}$, number of iterations $N_{\text{iter}}$, data dimension $d$
**Result:** whitening matrix $\mathbf{W}$
$\mathbf{W} \leftarrow \{0\}^{d \times d}$
**for** $i = 0; i < d; i{+}{+}$ **do**
    Sample $\mathbf{r} \sim U[-1, 1]^d$
    **for** $j = 0; j < N_{iter}; j{+}{+}$ **do**
        $\mathbf{r} \leftarrow \frac{\mathbf{Cr}}{\|\mathbf{Cr}\|}$
    **end**
    $\lambda \leftarrow \|\mathbf{Cr}\|$
    $\mathbf{C} \leftarrow \mathbf{C} - \lambda \mathbf{r}\mathbf{r}^T$
    $\mathbf{w}_{i\cdot} = \frac{1}{\sqrt{\lambda}}\mathbf{r}^T$
**end**
**return** $\mathbf{W}$

**Algorithm 1:** Constructing $\mathbf{W}$ by power iteration

---

Note that in algorithm 1 each operation is differentiable with respect to $\mathbf{C}$.

## 5 GRADIENT-BASED SLOW FEATURE ANALYSIS

The key idea for gradient-based SFA is that a whitening layer can be applied to any differentiable architecture (such as deep neural networks) to enforce outputs that approximately obey the SFA constraints while the architecture stays differentiable. As such, it can be trained using gradient-descent-like training procedures, allowing for hierarchical architectures, where every parameter is modified iteratively towards optimizing a **global** slowness objective, as opposed to assuming a local-to-global slowness as in HSFA. To formalize, if

$$\tilde{g}_\theta : \quad \mathbb{R}^{N \times d} \rightarrow \mathbb{R}^{N \times e}$$

is a differentiable function approximator, such as a neural network, parameterized by $\theta$ and

$$\mathcal{W}: \quad \mathbb{R}^{N \times e} \to \mathbb{R}^{N \times e}$$

denotes the approximate whitening procedure, then

$$g_\theta = \mathcal{W} \circ \tilde{g}_\theta: \quad \mathbb{R}^{N \times d} \to \mathbb{R}^{N \times e}$$

is an approximator whose outputs approximately obey the SFA constraints.

It is straightforward to define a general loss function as

$$L_\theta(\mathbf{X}) = \frac{1}{N} \sum_i \sum_j w_{ij} \|g_i - g_j\|^2 \qquad (2)$$

with $g_i$ being the $i$-th row of $g_\theta(\mathbf{X})$ and $w_{ij}$ being the strength of the connection between two points $x_i$ and $x_j$ similar to weights in spectral graph embeddings (cf. Sprekeler (2011) and Escalante-B. & Wiskott (2013)).

For optimizing slowness, we define the weight as

$$w_{ij} = \delta_{i,j+1}$$

with $\delta$ being the Kronecker delta. This connects consecutive steps in the time-series and disconnects the others [1].

The approximator can then be trained to minimize eq. 2 by following the negative gradient estimate of $L$ with respect to $\theta$, $-\nabla_\theta L$ for (mini-)batches of data, as common in a common in gradient-based training. In our experiments, we used the ADAM optimizer (Kingma & Ba (2014)) with Nesterov-accelerated momentum (Dozat (2015)) to train the approximator. Learning rate and additional hyperparameters were left at default values as implemented in the popular *Keras*-package (Chollet et al. (2015)).

## 6 EXPERIMENTS

### 6.1 SYNTHETIC TRIGONOMETRIC DATA

To show the general feasibility of this approach, we first show that PowerSFA finds near-optimal solutions in the linear case for synthetic data. Standard SFA is a linear method that relies on **(a)** non-linear basis function expansion and **(b)** hierarchical processing to induce non-linearity. This means that PowerSFA could hypothetically be used a similar fashion (even though a gradient-based approach allows for more complex models in a natural way).

The data is generated by trigonometric polynomials of degree $N$ as:

$$\mathbf{x}(t) = \boldsymbol{\varepsilon}_t + \sum_{n=1}^N \boldsymbol{\alpha}_n \cos(nt) + \boldsymbol{\beta}_n \sin(nt)$$

with $\mathbf{x}, \boldsymbol{\varepsilon}, \boldsymbol{\alpha}, \boldsymbol{\beta} \in \mathbb{R}^D$, coefficients $\alpha_{in}, \beta_{in} \sim \mathcal{N}(0, 1)$. A noise term $\varepsilon_{it} \sim \mathcal{N}(0, 0.01)$ is added to avoid numerical instabilities in the implementation of closed-form SFA, as singular covariance matrices can cause the underlying eigendecomposition to break.

We implemented a temporal step-size of $\frac{2\pi}{100}$, and generate $T = 5000$ steps with maximum degree $N = 30$ and dimension $D = 100$. The data is whitened with $N_{\text{iter}} = 60$.

Figure 1 shows the extracted features for different variants of SFA. If the slowness loss is optimized without any constraints on the output features, the optimal solution is to collapse all signals to a constant ($\Delta = 0$), while if unit variance is enforced, only the features with slowness very close to the smallest $\Delta$ are extracted multiple times and thus the representation becomes highly redundant. When using the approximate whitening, the quality of the solutions is comparable to the closed

---

[1]Note that this is not exactly the same as the SFA objective as it does not include an ordering of the features, but instead might rather be called a *slow subspace loss*. Any rotation of the data obeys the constraints and any optimal solution is just a rotated version of the solution of an ordered loss.

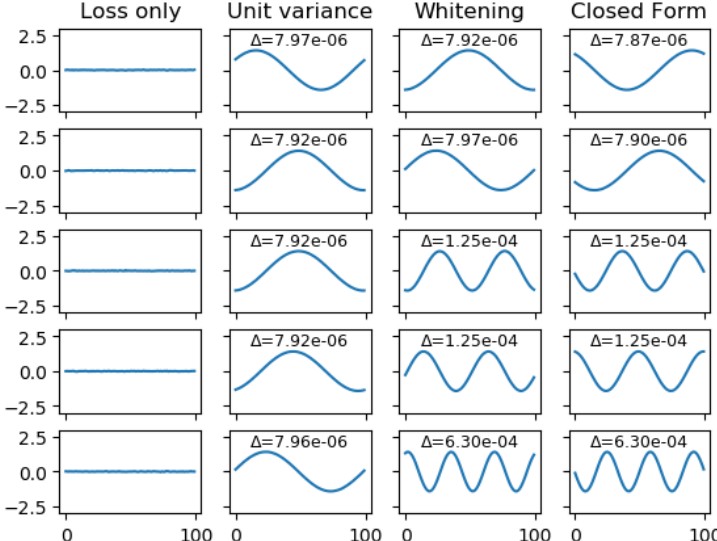

Figure 1: First five features (100 steps) learned by a linear model in different settings. From left to right: (1) only slowness loss optimized, no constraints, (2) slowness loss with enforced unit variance, (3) slowness loss with enforced whitening, (4) closed form solution. The $\Delta$-values indicate slowness (lower is slower).

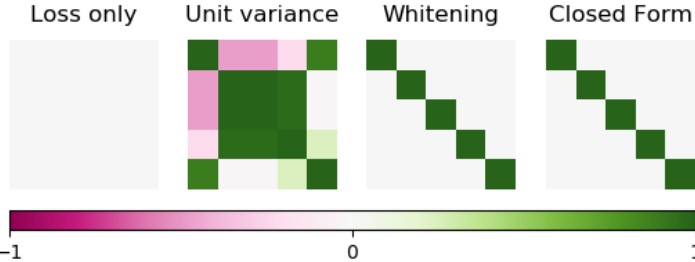

Figure 2: Covariance matrices for five outputs in different settings. (1) slowness loss without constraints leads to nearly constant signals with (co)variances close to zero, (2) unit variance enforced leads to highly correlated output signals, (3) & (4) (approximate) whitening and the closed form solution show fully decorrelated output signals (identity covariance).

form solution gained by solving a generalized eigenvalue problem. The ordering was achieved for representational purposes by weighting the output features with a monotonically decreasing weight. In Figure 2, the covariance matrices of the extracted signals are visualized.

Note that it is not a sensible approach to use PowerSFA to optimize a linear model for such low-dimensional data as the closed-form solution is easily attainable. For this reason, this experiment should be understood as a proof-of-concept and a general demonstration of applicability rather than a recommendation for a use-case.

A very low number of iterations will render the optimization unstable resulting in fast output signals. There seems to be no continuous trade-off between correlation and slowness mediated by the number of iterations. Thus, it is recommended to find a setting for $N_{\text{iter}}$ that allows for stable optimization. Appendix A contains a small hyper-parameter study, illustrating that behavior for the linear case.

To provide evidence on how gradient-based SFA can improve on solutions found by closed-form SFA, we encapsulate the original signal in a non-linear distortion:

$$\mathbf{u}(t) = \cos(e^{\mathbf{x}(t)})$$

This makes it impossible to extract slow signals by linearly unmixing the original components. To make the extraction more difficult, the maximum degree was increased to $N = 300$ and the temporal step-size was decreased to $\frac{2\pi}{10^5}$ to reduce temporal aliasing of very fast input components. Consequently, we increased $T$ to $10^5$ and $N_{\text{iter}}$ to 100.

When applying closed-form SFA to non-linear problems, it is common to apply multiple low-degree polynomial expansions interlaced with linear SFA steps, to reduce the dimensionality, in a greedy layer-wise training. We use an architecture with three quadratic expansion layers (normalized to unit norm to avoid exploding gradients), and compare **(a)** closed-form training and **(b)** gradient-based training. We repeat the comparison for a multi-layer perceptron with *tanh* activation, since polynomial expansion functions are uncommon in gradient-trained models. Both architectures are provided in Appendix B. Table 1 shows the average output slowness, once for the raw output from the network (with small residue correlations possible due to whitening approximation error) and once with closed-form whitening applied to precisely enforce the constraints for the most meaningful comparison.

| **Quadratic expansion** | Closed-form | Gradient-based |
|---|---|---|
| Slowness | $3.435 \cdot 10^{-1} \pm 3.54 \cdot 10^{-3}$ | $9.744 \cdot 10^{-2} \pm 1.67 \cdot 10^{-2}$ |
| Slowness (additional whitening) | $3.435 \cdot 10^{-1} \pm 3.54 \cdot 10^{-3}$ | $9.743 \cdot 10^{-2} \pm 1.67 \cdot 10^{-2}$ |

| **Neural network (tanh)** | Closed-form | Gradient-based |
|---|---|---|
| Slowness | $8.562 \cdot 10^{-1} \pm 6.79 \cdot 10^{-3}$ | $2.755 \cdot 10^{-1} \pm 9.891 \cdot 10^{-2}$ |
| Slowness (additional whitening) | $8.562 \cdot 10^{-1} \pm 6.79 \cdot 10^{-3}$ | $2.755 \cdot 10^{-1} \pm 9.891 \cdot 10^{-2}$ |

Table 1: Average slowness of 5 output features over 5 runs extracted by greedy layer-wise training and gradient-based training from non-linearly distorted trigonometric polynomials. Results for three-layer quadratic expansion network and neural network with *tanh* activation.

The results show that gradient-based training can improve slowness in multi-layer architectures compared to greedy layer-wise SFA. Note that increasing the output dimension of the dimensionality reduction steps (or dropping them altogether) in the quadratic expansion network will, unsurprisingly, lead to improved performance and ultimately to convergence to similar minima in both networks. However, this performance increase comes at the cost of high memory requirements and is usually not applicable in high-dimensional non-synthetic problems.

## 6.2 Pose Estimation from Visual Data

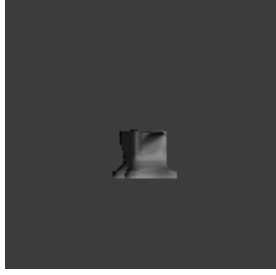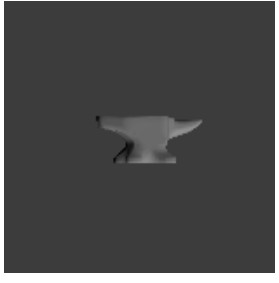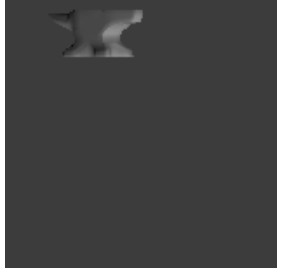

Figure 3: Three samples from the *anvil*-dataset in different $x, y$ positions and rotational angle $\Phi$.

SFA can be used to extract slowly varying underlying causes from high-dimensional data, such as object position, identity, and rotation from visual simulations. In Franzius et al. (2011), textured three-dimensional fish-objects that change in $x$- and $y$-position as well as in-depth rotation $\phi$ have been used and it has been shown that HSFA features encode position and angle well. Since the code used to generate the stimulus data was outdated and could not be executed anymore, we re-implemented a similar scenario with a 3D model of an anvil (Figure 3, Pino4et (2016)) using the random walk procedure described in Franzius et al. (2011) to generate the configurations.

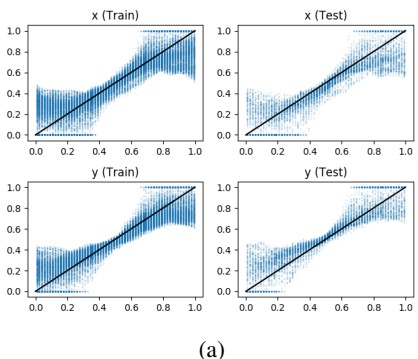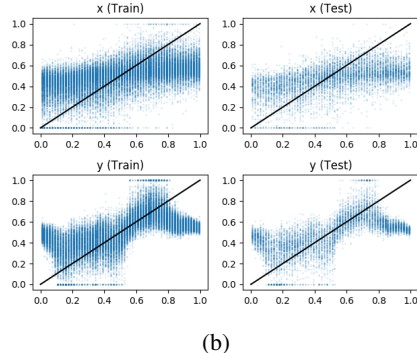

(a)                                                        (b)

Figure 4: Spread of the predicted $x$- and $y$-position of the *anvil*-dataset for a trained network (4a) and a randomly initialized network (4b). RMSE for the trained network on unseen data are $9\%$ and $8\%$ for $x$- and $y$- position respectively. RMSE for the random network on unseen data are $21\%$ and $21\%$.

A current limitation of our model (cf. section 7) is that it does not scale well in the number of output features, rendering the relatively high number of $512$ output features used by Franzius et al. (2011) infeasible. We used an output dimension of $10$ with a comparable architecture that replaces quadratic expansion by ELU-nonlinearities (a *Keras*-summary is given in Appendix C). Due to the low number of output features, we were not able to successfully extract the rotational angle $\Phi$.

In line with the original publication, we computed linear regression to predict $cos(\pi x)$ and $cos(\pi y)$ and used the inverse transformation to extract the object position. The results are shown in Figure 4 and are comparable to those presented by Franzius et al. (2011) as they were able to achieve a RMSE of $9\%$ for $x$-position and $7\%$ for $y$-position.

## 6.3   NORB

While gradient-based SFA is the main contribution of this work, previous work on generalizing SFA (Sprekeler (2011)) has shown that the SFA optimization problem is strongly related to a special case of the problem solved by a more general spectral embedding method, i.e., Laplacian Eigenmaps (Belkin & Niyogi (2003)), or, with small differences, graph-based SFA (Escalante-B. & Wiskott (2013)). For this reason, we defined a more general loss function in equation 2.

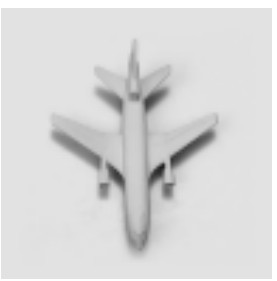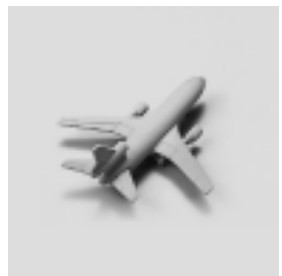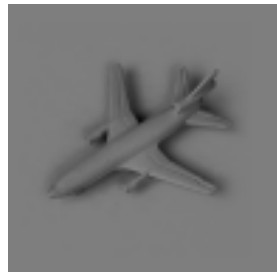

Figure 5: The embedded object from the NORB dataset. Samples differ in azimuth, elevation and lighting.

In fact, weights $w_{ij}$ can be defined between any two points $x_i$ and $x_j$ of a given dataset, not just consecutive ones, thus allowing to optimize neighborhood-respecting embeddings for general graphs. This is similar (but not fully equivalent) to spectral embeddings on graph data, as used in algorithms such as Laplacian eigenmaps. Our approach exhibits four significant differences:

1. It does not find ordered features, but a rotation of an optimal embedding,

2. the found solution should be assumed to be only locally optimal (or a saddle point),

3. it does ignore the graph (ir)regularity when enforcing the orthogonality constraint, and

4. it allows a natural and scalable way to embed unseen points.

Computing an out-of-sample embedding requires the same forward-pass through the differentiable architecture as the training embeddings. In particular, its complexity does not scale with the number of observed points used for training as is usually the case in other approximations, such as Nyström approximation (Williams & Seeger (2001)).

We demonstrate the usefulness of such an approach by embedding an object of the NORB dataset, a collection of photographs of toys taken at different elevations, azimuths, and under different lighting conditions with the *MobileNet* architecture (Sandler et al. (2018)) scaled with $\alpha = 0.5$ and a depth multiplier of 2 in the *Keras* implementation. Following Hadsell et al. (2006), we embedded photographs of a toy plane (Figure 5) in 972 configurations (18 azimuths $\times$ 6 lighting conditions $\times$ 9 elevations angles) that were randomly split into a train- and test-set of sizes 660 and 312 respectively and the connection weights $w_{ij}$ were chosen as 1 if $x_i$ and $x_j$ differed only in one step either in rotation (i.e., one azimuth) or elevation (i.e., one level) and 0 otherwise. The weights were independent of lighting condition.

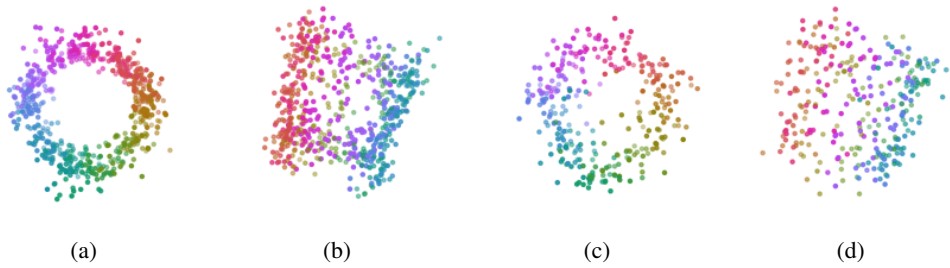

|        |        |        |        |
|--------|--------|--------|--------|
| (a)    | (b)    | (c)    | (d)    |

Figure 6: Cylindrical embedding of NORB plane with azimuth colored. 6a and 6b show the embedded training data from the front and the side of the cylinder respectively, while 6c and 6d show the test data for the same configuration. Circumference of the cylinder encodes the rotation of the plane.

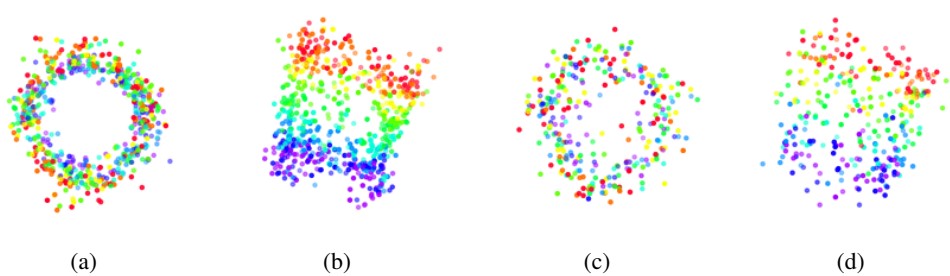

|        |        |        |        |
|--------|--------|--------|--------|
| (a)    | (b)    | (c)    | (d)    |

Figure 7: Cylindrical embedding of NORB plane with elevation colored. 7a and 7b show the embedded training data from the front and the side of the cylinder respectively, while 7c and 7d show the test data for the same configuration. Height on the cylinder encodes the photograph's elevation angle.

Figures 6 and 7 show the three-dimensional embedding that was found in this setting. The data was embedded in a cylindrical shape in which the circumference encodes the rotation angle of the embedded object, and the length along the cylinder encodes the elevation configuration of the object for the train-set and in the out-of-sample case of the test-set. Hadsell et al. (2006) found a similar cylindrical encoding, however, their results exhibit a more clean-cut embedding. We assume that this is due to DrLIM's maximization of distance for dissimilar samples that our model does not implement.

## 7 DISCUSSION

We propose a new way of extracting informative slow features from quickly varying inputs based on differentiable whitening of processed batches of the input data. To experimentally show the feasibility of the method, we trained a linear and two non-linear models to extract slowly varying output signals from synthetic time-series by gradient-descent, and find that the differentiable whitening ensures informativeness of the extracted features when optimizing a slowness loss function. Furthermore, the features corresponded to those found by closed-form SFA. [2]

To show applicability to visual time-series data, we trained a convolutional neural network on an input stream of an object randomly rotating in-depth and moving in a 2-dimensional plane. Gradient-based SFA preserves the position of the object in a low-dimensional representation, as does a hierarchical version of closed-form SFA. The rotation was not extracted successfully due to computational constraints in the naive implementation for a large number of output features (cf. 7).

One experiment was conducted on a non-time-series image dataset on which symmetric similarity relations between the data are defined. We show that a generalization of gradient-based SFA in the spirit of graph-based SFA is able to extract a low-dimensional representation that preserves and disentangles the configuration parameters used to define the similarity, in this case, azimuth and elevation of a photographed toy. This representation generalizes well to previously unseen configurations of the object.

While the algorithm is still in a prototypical state, the proof-of-concept results presented in this paper show promise for gradient-based SFA by differentiable whitening to extract meaningful representations for goal-oriented learning while leveraging the expressive power of modern architectures, such as convolutional neural networks. In particular, differentiable whitening ensures non-redundancy of the output features.

More research has to be dedicated to explore the computational limitations of this method and possibly lower the complexity for a larger number of output features. Furthermore, closed-form SFA has a strong theoretical framework describing the optimal responses in an idealized setting. At this point, we are unclear how well this framework translates to the proposed method of slowness extraction as the used models typically suffer from local optima when being iteratively trained.

One limitation of PowerSFA is that it currently does not scale favorably in the number of output features $e$. We see two main reasons for this: the necessary batch size to get a meaningful estimate of the batch-covariance estimate and its calculation. The latter is due to the complexity of a naive $\mathbb{R}^{e \times N_{\text{batch}}} \cdot \mathbb{R}^{N_{\text{batch}} \times e}$ matrix-multiplication being $\mathcal{O}(N_{\text{batch}} e^2)$, while the former is due to the whitening procedure expecting a covariance matrix of full rank and thus $N_{\text{batch}} \geq e$ samples.

To reduce the lower bound on the batch-size at batch $t$, a convex mixture with the covariance matrix of the previous batch might be applicable:

$$\mathbf{C}_t = (1 - \gamma)\mathbf{C}_\theta + \gamma\mathbf{C}_{t-1}$$

Note, that only the current batch's covariance matrix $\mathbf{C}_\theta$ is considered parameter-dependent and allows to propagate a gradient for training. Thus, large values for $\gamma$ might cause a significant bias in the gradient-estimate. This has not been used to generate the proof-of-concept results for this paper.

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

# Appendices

## A   SMALL PARAMETER-STUDY: NUMBER OF POWER ITERATIONS

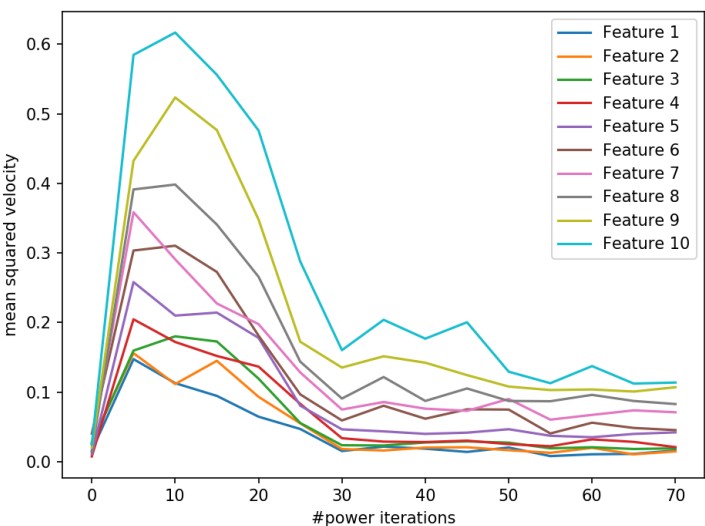

Figure 8: The mean squared velocity (unnormalized $\Delta$ value) per feature for an increasing number of power iterations, averaged over 10 trials. For no power iterations, the velocity is close to 0, but for a too small positive number of power iterations the optimization becomes unstable and results in sub-optimal performance.

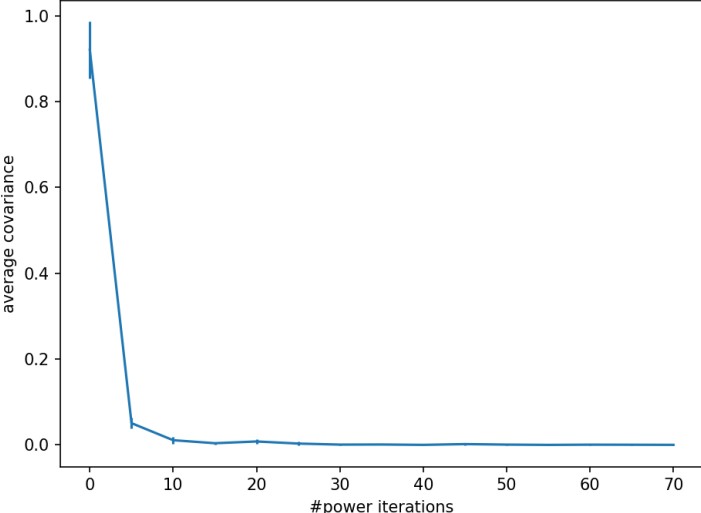

Figure 9: The average covariance (auto-variances not included) for an increasing number of power iterations, averaged over 10 trials. Without whitening, the covariance is close to 1, but the features quickly become decorrelated when the number of iterations is increased.

## B  COMPARISON ARCHITECTURES (SYNTHETIC DATA)

This appendix presents the architectures used for comparing greedy layer-wise with gradient-based end-to-end training of SFA. Both networks have been trained with both training methods. In the gradient-based setting, the *Keras*'s *Nadam* optimizer has been used with default hyperparameters and a whitening layer has been applied to the network output.

| Quadratic expansion network | |
| --- | --- |
| Operation | Output dimension |
| Input Layer | 100 |
| Linear Layer | 33 |
| Quadratic Expansion | 594 |
| Linear Layer | 33 |
| Quadratic Expansion | 594 |
| Linear Layer | 33 |
| Quadratic Expansion | 594 |
| Linear Layer | 5 |

Table 2: A network with multiple quadratic expansions, each preceded by linear dimensionality-reduction. These kind of networks are typically used in closed-form SFA as they trade-off model expressivity with memory requirements.

| Neural network(tanh) | |
| --- | --- |
| Operation | Output dimension |
| Input Layer | 100 |
| Linear Layer | 100 |
| Pointwise *tanh* | 100 |
| Linear Layer | 100 |
| Pointwise *tanh* | 100 |
| Linear Layer | 100 |
| Pointwise *tanh* | 100 |
| Linear Layer | 5 |

Table 3: A simple multi-layer neural network using a *tanh* activation function to induce non-linearity.

## C  KERAS DESCRIPTION (HSFA-LIKE ARCHITECTURE)

```
 Layer (type)                 Output Shape              Param #
=================================================================
input_1 (InputLayer)         (None, 156, 156, 1)       0
_________________________________________________________________
conv2d_1 (Conv2D)            (None, 147, 147, 32)      3232
_________________________________________________________________
activation_1 (Activation)    (None, 147, 147, 32)      0
_________________________________________________________________
dropout_1 (Dropout)          (None, 147, 147, 32)      0
_________________________________________________________________
conv2d_2 (Conv2D)            (None, 28, 28, 32)        102432
_________________________________________________________________
activation_2 (Activation)    (None, 28, 28, 32)        0
_________________________________________________________________
dropout_2 (Dropout)          (None, 28, 28, 32)        0
_________________________________________________________________
conv2d_3 (Conv2D)            (None, 25, 25, 32)        16416
_________________________________________________________________
activation_3 (Activation)    (None, 25, 25, 32)        0
_________________________________________________________________
dropout_3 (Dropout)          (None, 25, 25, 32)        0
_________________________________________________________________
conv2d_4 (Conv2D)            (None, 11, 11, 32)        16416
_________________________________________________________________
activation_4 (Activation)    (None, 11, 11, 32)        0
_________________________________________________________________
dropout_4 (Dropout)          (None, 11, 11, 32)        0
_________________________________________________________________
conv2d_5 (Conv2D)            (None, 8, 8, 32)          16416
_________________________________________________________________
activation_5 (Activation)    (None, 8, 8, 32)          0
_________________________________________________________________
dropout_5 (Dropout)          (None, 8, 8, 32)          0
_________________________________________________________________
conv2d_6 (Conv2D)            (None, 3, 3, 32)          16416
_________________________________________________________________
activation_6 (Activation)    (None, 3, 3, 32)          0
_________________________________________________________________
dropout_6 (Dropout)          (None, 3, 3, 32)          0
_________________________________________________________________
flatten_1 (Flatten)          (None, 288)               0
_________________________________________________________________
dense_1 (Dense)              (None, 10)                2890
_________________________________________________________________
power_whitening_1 (PowerWhit (None, 10)                0
=================================================================
Total params: 174,218
Trainable params: 174,218
Non-trainable params: 0
_________________________________________________________________
```

