# OpenReview forum: "Gradient-based Training of Slow Feature Analysis by Differentiable Approximate Whitening"
_ICLR.cc/2019/Conference_

### Official Review · AnonReviewer1 · 2018-10-29
**Neural implementation of approximate SFA with a whitening layer using power iterations.**

**Rating:** 6
**Confidence:** 4

**Review:**

Summary
The manuscript proposes to use power iterations in an approximate "whitening layer" to optimize the slowness objective of SFA in a very general setting. A set of experiments illustrates that this way of doing nonlinear SFA is meaningful.

Quality
Although the idea is pretty straight forward and the paper shows qualitative results on a number of datasets, the relative merit of the approach is empirically not well characterized.

Clarity
The manuscript is in general well written and the technical content is well accessible. However the description of the whitening layer implementation needs some more details.

Originality
The idea of using a whitening layer together with the slowness objective has not been explored before. There is a second ICLR 2019 submission (Pfau et al.) with a very similar idea, though.

Empirical Evaluation
The approximate whitening should lead to a trade-off between whitening and slowness optimization. I miss an experiment illustrating that trade-off. Also the comparison to nonlinear SFA using expansion or kernelization of hierarchical SFA is empirically not properly characterized. In the end, if one takes the slowness objective seriously, one would use the method yielding slower results.

Significance
The manuscript introduces a way of running nonlinear SFA with approximate constraints in a general deep learning setting with a differentiable implementation using a dedicated whitening layer based on power iterations.

Reproducibility
The data is either synthetic or publicly available. The Keras implementation of the PowerWhitening layer as well as the entire neural network along with its optimization schedule is not shared. Hence, there should be some effort involved to reproduce the experiments.

Pros and Cons
1+) The idea of an approximate whitening layer is conceptually simple and clear.
2-) The description of the practical implementation of the power iteration is slightly imprecise.
3-) The algorithm scales badly in the number of output dimensions. This scaling is bad in a computational sense and also in a statistical sense.

Details
a) Section 6.1: Why do you need to add the noise term? What is the statistical meaning of this added noise?
b) Section 6.1: the solutions if comparable -> the solutions is comparable
c) References: Shaham -> ICLR 2018 paper
d) References: nyström -> Nyström
e) The name for the algorithm "Power SFA" is a little bit bold.

---

> ### Author Response · Authors · 2018-11-27
> **Very helpful review**
>
> Dear Reviewer,
>
> we are very thankful for this detailed and actionable review, it gave us a lot to work with to, hopefully, improve our paper for the current revision.
>
> On Empirical Evaluation & Quality:
> We included an experiment on the synthetic dataset investigating the progression of average correlation and slowness for an increasing number of power iterations. Surprisingly, there seems to be no trade-off, but too low a number of power iterations will result in sub-optimal performance even if the average correlation is already low. We describe this qualitatively in the section on the experiments using synthetic data, and we included the progression plots for both metrics in the Appendix.
>
> Also, in Section 6.1, we conducted new experiments with non-linear models (a quadratic expansion network, common for non-linear SFA, and a multi-layer perceptron) on a non-linearly distorted version of the synthetic data. The experiments show that gradient-based training can improve performance when compared with greedy layer-wise training.
> Two things should be noted: (a) The experimental setup is not exhaustive (e.g., no systematic model selection has been conducted), but we believe it does illustrate the conceptual promise of the paradigm. (b) Kernelized SFA has not been included in these experiments, as there is no implementation readily available at the moment.
>
> The new architectures have been shared as a high-level overview in the Appendix and as actual implementation in the provided code.
>
>
> On Clarity & Reproducibility:
> To promote reproducibility, we uploaded code for the synthetic experiments (including the newly conducted ones) as well as for the NORB experiments. The "Discussion" section contains a link to the archive. The code includes a stand-alone implementation of the whitening layer in the Keras framework. We hope that this improves comprehensibility and encourages original experimentation with this method.
>
>
> On Significance:
> Nothing for us to reply to here, the description is spot-on.
>
>
> On Originality:
> We tried to make the differences between the mentioned work and ours more visible. While both papers are aiming to solve a similar problem, our method relies mainly on an simple architectural change (which is, however, applicable independently of the objective), while SpINs are theoretically derived directly from a more general problem formulation and their success stems from a smart optimization of a loss that enforces the constraints. We do see distinct merit in both approaches.
>
>
> On Details:
> a) The noise is a precautionary measure as closed-form SFA can have the problem of breaking due to linear dependencies in the data, and we do sample the data-generating coefficients randomly. While the current version of MDP (the framework we used for closed-form SFA) available on github includes methods to deal with this problem, the initial experiments have been conducted with the version published on PyPI which does not have this feature yet. We included a brief note on this in Section 6.1.
> b, c, d) Have been corrected.
> e) We agree that it is a little bold, but we decided to go with this name for two reasons: It hints at the core method underlying the algorithm and thus might be a useful mnemonic, and it caught on in our discussions, despite the actual decision to change it. As it is hard to come up with memorable and descriptive names, we ultimately decided to be a little bold.
>
> We greatly appreciate the time and effort you spent on reviewing our submission and hope you find your concerns addressed in our current revision.
>
> Respectfully,
> the Authors

---

### Official Review · AnonReviewer2 · 2018-11-04
**Gradient-based SFA**

**Rating:** 6
**Confidence:** 4

**Review:**

In this paper the authors present a differentiable objective for slow feature analysis, to facilitate end-to-end training.   I am not clear on the novelty of this formulation, as it appears to have been proposed in a similar form in previous works (e.g., A maximum-likelihood interpretation for slow feature analysis by Turner and Sahani - Eq., (2)) and can probably be considered straightforward.  Nevertheless, the approximate whitening layer and the way it is used is a smart approach for this problem.  The experiments are interesting and shed light on the properties of the method.  In summary, the paper may lack technical novelty in some respect, but the experiments are convincing in terms of proof-of-concept, and the approach is smart.

---

> ### Author Response · Authors · 2018-11-28
> **We tried to sharpen the contrast of the original parts of this contribution**
>
> Dear Reviewer,
>
> Your intuition is right, the objective function itself is not novel and is indeed straightforward to derive. In fact, it is just an unordered version of the original SFA loss (which is in itself differentiable already).
>
> The whole novelty lies in the idea of using the described whitening layer, thus, incorporating the constraints in the model rather than the loss function. The paper you mentioned can be seen as doing the latter. Thus, we understand it to fall in line with "Bergstra, J. and Bengio, Y. - Slow, decorrelated features for pretraining complex celllike networks" where a penalty term for covariances is included in the loss function. Futhermore, the probabilistic model by Turner and Sahani is not optimized by gradient-descent, but trained by the Expectation Maximization algorithm.
>
> We tried to clarify that the derivation of the objective function is not an involved or novel part of our algorithm, e.g., in the description for equation (2). With the introductory sentence to Section 5 ("The key idea for gradient-based SFA is that a whitening layer can be applied to any differentiable architecture [..] to enforce outputs that approximately obey the SFA constraints while the architecture stays differentiable."), we hope that this is helps to clearly pinpoint what the original parts of our work are.
>
> Thank you for taking the time and effort to review our paper!
> Hopefully, you find the core idea more unambigiously presented in the current revision.
>
> Respectfully,
> the Authors

---

### Official Review · AnonReviewer3 · 2018-11-12
**A natural place to incorporate deep learning**

**Rating:** 5
**Confidence:** 2

**Review:**

The authors state a clear summary of their contribution to Slow Feature Analysis (SFA) in Section 5: "The key idea for gradient-based SFA is that a whitening layer can be applied subsequently to any differentiable architecture (such as deep neural networks) to enforce outputs that approximately obey the SFA constraints while still being a differentiable architecture.” As a result, the proposed method can replace the previous SFA pipeline of [fixed, non-linear feature extraction + learned linear feature extraction] with simply end-to-end [learned, non-linear feature extraction]. The resulting method is thus more flexible and expressive and less hand-crafted than traditional SFA approaches. The paper shows experiments that validate that the method can learn meaningful representations in practice.

The writing is difficult to follow, unclear in several places, and has grammatical mistakes. As a result, it is more challenging to understand the proposed method, its motivation, and its precise place among related literature. For example, the authors discuss SpIN, and from that discussion, it seems that SpIN are quite similar; SpIN was submitted to arXiv 3-4 months before the ICLR deadline and thus is not concurrent as the authors claim (on my understanding, from reading the ICLR and other ML conference guidelines).

The approach does seem somewhat incremental, though I would be open to changing my mind on author response. On one view, this method can be seen as simply replacing SFA’s fixed, non-linear feature extraction with learned non-linear feature extraction, with a trick to make it work. Deep learning consistently improves over fixed non-linear feature extractors, so it is not surprising a surprising place to incorporate neural networks.

Based on the method, this paper could go either way, but given my concerns on its novelty and writing quality/clarity, I lean slightly towards reject.

---

### Meta-Review · Area_Chair1 · 2018-12-15
**Interesting approach but somewhat limited analysis**

**Confidence:** 4
**Recommendation:** Reject

**Metareview:**

This paper proposes to unroll power iterations within a Slow-Feature-Analysis learning objective in order to obtain a fully differentiable slow feature learning system. Experiments on several datasets are reported.

This is a borderline submissions, with reviewers torn between acceptance and rejection. They were generally positive about the clarity and simplicity of the presentation, whereas they raised concerns about the relative lack of novelty (especially related to the recent SpIN model), as well as the current limitations of the approach on large-scale problems. Reviewers also found authors to be responsive and diligent during the rebuttal phase. The AC agrees with this assessment, and therefore recommends rejection at this time, encouraging the authors to resubmit to the next conference cycle after addressing the above points.